# Optimization of Antibacterial Activity and Biosafety through Ultrashort Peptide/Cyclodextrin Inclusion Complexes

**DOI:** 10.3390/ijms241914801

**Published:** 2023-09-30

**Authors:** Hang Liu, Lin Wang, Chen Yao

**Affiliations:** School of Chemistry and Chemical Engineering, Southeast University, Nanjing 211189, China; 220213312@seu.edu.cn (H.L.); 220203166@seu.edu.cn (L.W.)

**Keywords:** antimicrobial peptide, β–cyclodextrin, inclusion complexes, antibacterial activity, biosafety

## Abstract

Engineered ultrashort peptides, serving as an alternative to natural antimicrobial peptides, offer benefits of simple and modifiable structures, as well as ease of assembly. Achieving excellent antibacterial performance and favorable biocompatibility through structural optimization remains essential for further applications. In this study, we assembled lipoic acid (LA)–modified tripeptide RWR (LA–RWR) with β–cyclodextrin (β–CD) to form nano–inclusion complexes. The free cationic tripeptide region in the nano–inclusion complex provided high antibacterial activity, while β–CD enhanced its biocompatibility. Compared with peptides (LA–RWR, LA–RWR–phenethylamine) alone, inclusion complexes exhibited lower minimum inhibitory concentrations/minimum bactericidal concentrations (MICs/MBCs) against typical Gram–negative/Gram–positive bacteria and fungi, along with improved planktonic killing kinetics and antibiofilm efficiency. The antibacterial mechanism of the nano–inclusion complexes was confirmed through depolarization experiments, outer membrane permeability experiments, and confocal laser scanning microscopy observations. Furthermore, biological evaluations indicated that the hemolysis rate of the inclusion complexes decreased to half or even lower at high concentrations, and cell viability was superior to that of the non–included peptides. Preliminary in vivo studies suggested that the inclusion complexes, optimized for antibacterial activity and biosafety, could be used as promising antibacterial agents for potential applications.

## 1. Introduction

Antimicrobial peptides (AMPs) are an inherent part of the biological immune system and exhibit broad–spectrum antibacterial activity against various pathogens, including bacteria, fungi, and even viruses. Due to their cationic and hydrophobic nature, AMPs possess a typical membrane–targeting mechanism that involves binding to and disrupting cell membranes [1]. This makes them highly effective against the multidrug–resistant strains, while reducing the probability of resistance development [2,3]. Despite their significant advantages, the structural complexity and low protease stability of AMPs limit their clinical applications, prompting studies on the development of engineered ultrashort peptides as alternatives.

The design of engineered peptides typically involves amino acid substitutions or truncations based on natural AMP sequences. Engineered cationic peptides often include multiple lysine or arginine residues to afford a net positive charge [4,5,6]. Additionally, the hydrophobic alkyl chain modification on the N–/C–terminus of these peptides is frequently required. It serves to enhance the hydrophobic interactions between the peptides and cell membranes, thereby exhibiting excellent antibacterial performance and good protease stability [7]. Myristic acid (14 carbons) or palmitic acid (16 carbons) has been usually employed to modify tripeptides/tetrapeptides as engineered lipopeptides [8,9]. Zhong et al. [10] reported that peptides conjugated with longer fatty acid chains (12 carbons) exhibited better antibacterial activity but lower selectivity. Higher selectivity indicated a greater preference for targeting anionic bacterial membranes over zwitterionic mammalian cell membranes. A moderate hydrophobic modification within the 8–10–carbon range was revealed to achieve optimal antibacterial performance and high selectivity. Almahboub et al. [11] conducted both the N– and C–terminal conjugation of an AMP derived from lactoferricin B with 2–aminooctanoic acid. The antibacterial performance of the modified peptides was enhanced by up to 16–fold, while peptides modified at the C–terminus exhibited lower MICs than those modified at the N–terminus. However, researchers observed significant cytotoxicity/hemolysis with peptides with long lipid chains, such as C16–KLLK and C14–KLLK. Jensen et al. [12] confirmed that the lipid length was the driving force for both high antibacterial and hemolytic activity. They suggested that the carbon–chain length for peptide modification should not exceeded 10. Therefore, achieving superior antibacterial performance, along with reduced side effects, is crucial for the further application of modified AMPs.

Our group has employed lipoic acid (LA), a natural antioxidant, to modify Bac8c (a derivative of the bactinecin linear variant Bac2A) and other tripeptides/tetrapeptides [13,14]. LA, characterized by a five–membered ring with disulfide bonds, can undergo polymerization to form polysulfide crosslinks and has extensive applications for constructing reduction–responsive reversible crosslinked nanoparticles [15,16]. Tripeptides modified with LA exhibited significantly enhanced antibacterial performance after crosslinking into nanoparticles [14].

Cyclodextrins (CDs) are naturally derived cyclic oligosaccharides commonly used for forming inclusion compounds (ICs) via host–guest interactions [17]. Various drugs have been complexed with natural or derived CDs to improve their physical and chemical properties, including enhanced solubility, stability, bioavailability, and reduced toxicity [18]. The complexation of HP–β–CD with 9–nitrocamptothecin reduced the drug–related toxicity and enhanced its anti–tumor efficacy [19]. The CD–Pt prodrug exhibited anticancer effects comparable to cisplatin at a single dose but with significantly reduced toxic side effects [20]. It is worth noting that LA is also well suited as a guest of CDs in water to obtain inclusion complexes with large binding constants [21]. Therefore, the inclusion of an LA moiety in the CD cavity, while leaving the cationic peptide region free, presents a worthwhile attempt with the potential to improve the stability and biocompatibility of AMPs.

In this study, engineered cationic tripeptides were modified by LA (LA–RWR, simplified as LR), followed by encapsulation via β–CD due to the host–guest interaction between the β–CD cavity and LA moiety. The inclusion nano–complexes were characterized using a UV–visible spectrophotometer, FT–IR, and TEM. The minimum inhibitory concentration (MIC), minimum bactericidal concentration (MBC), planktonic killing kinetics, and anti–biofilm activity of β–CD/LR and β–CD/LRP (LA–RWR–phenethylamine, simplified as LRP) nano–complexes were investigated in comparison with the modified tripeptides. The antibacterial mechanism was confirmed through membrane depolarization and permeability tests. The biocompatibility of the inclusion complexes was evaluated via hemolysis and cytotoxicity assays. Furthermore, a preliminary in vivo study was conducted to demonstrate the bactericidal effect and biosafety of the inclusion nano–complexes.

## 2. Results

### 2.1. Characterization of β–CD/LR and β–CD/LRP Inclusion Complexes

The UV spectra of the inclusion complexes indicated a decrease in peptide concentrations before and after inclusion at different inclusion ratios (see Figure 1a). Compared with the IR data of LA, the S–S characteristic peak at 671 cm^−1^ and the C–S characteristic peak at 734 cm^−1^ disappeared in those of β–CD/LR and β–CD/LRP [22], respectively (see Figure 1b). In addition, the characteristic peaks of β–CD/LR and β–CD/LRP were similar to that of β–CD with a decreasing trend in the layers from 3000 cm^−1^ to 3500 cm^−1^, indicating the formation of inclusion complexes. The morphological characteristics of nanostructured β–CD/LRP inclusion complexes were confirmed via TEM, which showed that the average diameter was about 120 nm (see Figure 1c). As is often observed in concentrated complexation media [23], the self–aggregation of CD/peptide complexes and free peptides can lead to an increased local density of peptides, resulting in enhanced performance. The most common type of CD/peptide complex has a 1/1 inclusion ratio. CD/peptide complexes can readily reform complex aggregates with CDs [24]. In this study, it was found that β–CD/peptide complexes with inclusion ratios between 1/2 and 1/5 exhibited excellent antibacterial performance, with the best performance observed at 1/5. This can be attributed to the possibility that the nano–inclusion complexes may induce outer membrane/lipopolysaccharide destabilization, while free peptides could transit across the outer membrane easily via self–promoted uptake. Therefore, the inclusion ratio of the β–CD/LR complex used for subsequent discussion was 1/5, while that of β–CD/LRP complex was 1/3.

### 2.2. In Vitro Antibacterial Activity

The minimum inhibitory concentrations (MICs) and minimum bactericidal concentrations (MBCs) of the inclusion complexes against representative Gram–negative bacteria, Gram–positive bacteria, a drug–resistant *MRSA* strain, and yeast are listed in Table 1. After encapsulation with β–CD, there was an approximate 32–fold reduction in both the MICs and MBCs of LR towards *E. coli* and *S. aureus*. Specifically, the β–CD/LR inclusion complexes exhibited an MIC value of 8 μg/mL for *MRSA*, while that of LR was 256 μg/mL. The MIC of β–CD/LR for *C. albicans* also decreased from 128 μg/mL to 8 μg/mL after encapsulation. The results indicated that the antibacterial activity of LR was significantly enhanced via β–CD encapsulation. On the other hand, phenylethylamine modification significantly improved the antibacterial activity of LR according to the MICs and MBCs of LRP. Further encapsulation of LRP reduced the MIC and MBC to half of their original values, which were closest to those of the commercial daptomycin.

The antibacterial stability of β–CD/LR and β–CD/LRP was confirmed by comparing its MICs under different conditions as listed in Table 2. The results demonstrated that the antibacterial activity of the inclusion complexes against *E. coli* was almost independent of the pH and time, with a slight decrease in the presence of salt or at higher serum concentrations (20%).

### 2.3. Planktonic Killing Kinetics

The antibacterial kinetics of the inclusion complexes were assessed by plotting the logarithmic values of viable bacterial cell numbers over various time intervals. As for *E. coli*, LR exhibited a logarithmic reduction from 6.4 to 4.5 within 1 h, and a reduction of approximately three orders of magnitude within 4 h. Under equivalent conditions, β–CD/LR showed a reduction of approximately one order of magnitude within 10 min, which meant that about 90% of the cells were killed in such a short time. Within 2 h, the viable bacterial numbers decreased by 4.4 magnitude orders. The results indicated a superior performance of LR after β–CD encapsulation. Additionally, β–CD/LRP showed a slightly higher bactericidal rate than β–CD/LR (approximately 0.4 magnitude order) at the same time point, but there was no significant difference when compared with LRP. Similar results were found with the other bacterial strains, as shown in Figure 2b–d.

### 2.4. Anti–Biofilm Experiment

The effect of inclusion complexes on preformed bacterial biofilms was assessed using crystal violet staining. Figure 3 demonstrates a significant reduction in the biofilm density of *E. coli* after treatment with β–CD/LR, resulting in a decrease to 19%. This reduction was notably lower compared to that with LR alone, which yielded a reduction in the biofilm density of 48%. Furthermore, the phenylethylamine modification of LR led to enhanced anti–biofilm performance, while the most remarkable results were observed with β–CD/LRP inclusion complexes. β–CD/LRP achieved a biofilm removal rate of 87.7%, comparable to that of the commercially available daptomycin. The results also indicated that β–CD/LR and β–CD/LRP exhibited similar effects on the preformed biofilms of *S. aureus*, *MRSA*, and *C. albicans*.

### 2.5. Visualization of Cell Damage

The morphology of *E. coli* and *C. albicans* incubated with inclusion nano–complexes was observed using confocal laser scanning microscopy (CLSM). SYTO 9 was utilized to stain live bacteria, emitting green fluorescence and capable of permeating all cell membranes. Propidium iodide (PI) was used to stain dead bacteria (red fluorescence) and penetrated membranes with permeability changes. As shown in Figure 4, bacterial cells treated with LR showed only faint red fluorescence, suggesting that the cell membranes were not significantly affected. In contrast, bacterial cells treated with LRP, β–CD/LR, and β–CD/LRP exhibited intense red staining, indicating substantial disruption of and damage to the cell membranes.

### 2.6. Antibacterial Mechanisms of Peptides

Our previous studies had demonstrated that the cationic peptides LR and LRP exhibit antibacterial activity by targeting bacterial cell membranes. In order to confirm the membrane–related mechanism of the inclusion complexes, the effects on the bacterial membrane potential and outer membrane permeability were investigated. According to the literature, cationic antimicrobial peptides interact with bacterial cell membranes, leading to membrane depolarization and the formation of ion channels [25].

The depolarization can be visualized based on the release of the fluorescent probe diSC_3_5 into the buffer solution. Changes in fluorescence intensity can demonstrate the ability of inclusion complexes to disrupt the bacterial cell membrane potential. Figure 5a illustrates that the fluorescence intensity of *E. coli* DC2 suspensions, upon adding inclusion complexes, increased dramatically within one minute, indicating the swift disruption of the bacterial membrane potential. Both β–CD/LR and β–CD/LRP complexes exhibited similar depolarization behavior. However, the fluorescence intensity of β–CD/LRP showed a slight decline after one minute. This may be attributed to the hindrance of diSC_3_5 probe entry by the outer membrane of Gram–negative bacteria. As the complexes become more effective, facilitating outer membrane permeabilization, this enables the probe to be inserted into the polarized cytoplasmic membrane, consequently leading to a reduction in its fluorescence.

The fluorescent probe NPN was used to investigate the effects of β–CD/LR and β–CD/LRP on bacterial outer membrane permeability. Once the inclusion complexes exert a disruptive effect on the outer membrane, NPN is allowed to penetrate the hydrophobic environment between the lipid bilayers of the outer membrane, thus generating robust fluorescence. As depicted in Figure 5b, within one minute following the addition of β–CD/LR and β–CD/LRP, the fluorescence intensity increased rapidly, indicating a potent influence on the outer membrane. 

These findings indicated that the inclusion complexes, like the cationic peptides, followed a mechanism of action that targets the bacterial cell membrane.

### 2.7. In Vitro Biosafety

An in vitro biosafety assessment, including the hemolysis rate and cell viability, was conducted on LR, LRP, and the inclusion complexes β–CD/LR and β–CD/LRP.

As listed in Table 3, the hemolysis rate of the inclusion complexes at the same concentration was approximately 50% lower than that of the respective peptides. The hemolysis rate of LR and LRP exceeded 5% at high concentrations (512 μg/mL), whereas that of β–CD/LR and β–CD/LRP remained below 2.5%. As LRP exhibited a higher hemolysis rate compared with LR due to phenylethylamine modification, the best hemolytic performance was observed for β–CD/LR. Notably, both the inclusion complexes showed superior hemolytic performance to commercial daptomycin. 

Figure 6 presents the cytotoxicity of the peptides and their inclusion complexes with concentrations ranging from 2 to 512 μg/mL. At a concentration of 128 μg/mL, the cell viability with LR significantly declined to 75%, whereas the cell viability with β–CD/LR was maintained at 96%. After phenylethylamine modification, the cell survival rate was less than 80% at an LRP concentration of 8 μg/mL, indicating obvious cytotoxicity. After encapsulation with β–CD, the cytotoxicity of LRP was greatly reduced. When the concentration of β–CD/LRP was below 64 μg/mL, the cell survival rate was maintained above 97%. As the concentration increased to 128 μg/mL, the cell survival rate with β–CD/LRP was 85%, while that with LRP alone was below 54%.

The results suggested that the encapsulation of the peptides with β–CD significantly improved the hemolytic performance and cytotoxicity of the peptides significantly at high concentrations.

### 2.8. In Vivo Antibacterial Performance

Here, 300 μL of an *MRSA* suspension at a concentration of 1 × 10^8^ CFU/mL was intravenously injected into the tail veins of mice, establishing an infectious model. After a period of 24 h, 100 μL of murine blood was spread onto the agar plate to identify the *MRSA* concentration as being 3.2 × 10^6^ CFU/mL. The mice infected with *MRSA* were treated with the peptides and their inclusion complexes. The logarithmic values of the viable bacterial concentrations in blood at various time intervals were plotted, as illustrated in Figure 7. The results showed no significant difference in bactericidal kinetics between LR and its encapsulated complex, both displaying a decrease in the bacterial concentration by approximately three orders of magnitude within 6 h. In comparison, the in vivo bactericidal activity of LRP and its complex was superior within the same 6 h duration, demonstrating a reduction in the bacterial concentration by approximately 4.3 orders of magnitude (that is a bactericidal efficiency exceeding 99.99%). As shown in Figure 8, after 24 h of treatment with β–CD/LRP, the bacterial concentration in murine blood was decreased to merely 15 CFU/mL. The results suggested that the encapsulation of LRP with β–CD, which improved the biosafety, also retained excellent performance in terms of the antibacterial activity in vivo.

As is shown in Figure 8a, compared with the positive control group, liver tissues 24 h after inclusion injection showed a clear cloacal structure, no obvious necrosis, and lymphocyte and inflammatory cell infiltration. For the spleen tissue, the experimental group showed no significant overall abnormalities, except for a few neutrophil infiltrations (see Figure 8b). The results indicated that no major inflammatory response occurs after the injection of the inclusion complexes.

## 3. Discussion

In recent years, engineered AMPs were developed based on the use of arginine/lysine as cationic amino acids and other hydrophobic amino acids as supplements. The design involved incorporating a hydrophobic tail to compensate for the natural hydrophobic amino acids within the AMP chain, resulting in outstanding antibacterial performance [26]. It has been reported that the antibacterial activity of AMPs decreases as the binding fatty acid tails are removed. The hydrophobic interactions conferred by the fatty acids enhance the peptide’s affinity for the bacterial membrane, thus enhancing the activity of AMPs. We have modified the tripeptides RWR with LA instead of fatty acids, and they exhibited moderate antibacterial activity (e.g., MIC of 128 μg/mL against *E. coli*) [14]. The further crosslinking of LA to form c–LA–RWR nanoparticles significantly enhances their antibacterial activity (e.g., MIC of 4 μg/mL against *E. coli*). These results suggested that besides the interaction between the hydrophobic region of the peptide chain and the lipid layer of the bacterial membrane, the impact of nanoparticles on membrane permeability should not be overlooked [27].

Based on the host–guest interaction between the LA moiety and β–CD cavity, we assembled the LA–modified cationic tripeptide RWR (LR) with β–CD to form nano–inclusion complexes in the present study. The β–CD/LR inclusion complexes exhibited higher antibacterial activity compared to LR alone (e.g., MIC of 4 μg/mL against *E. coli*). The stability of the inclusion complexes has also been confirmed (Table 1). LR modified with benzylamine at the C–terminus possesses improved antibacterial activity, which is further enhanced by β–CD inclusion. Both β–CD/LR and β–CD/LRP inclusion complexes exhibit excellent bactericidal effects against drug–resistant bacteria (*MRSA*) and bacterial biofilm.

Depolarization and outer membrane permeability studies confirmed that the inclusion complexes exhibited the typical mechanism of action similar to AMPs, including membrane destabilization and disruption. As the LA moiety being encapsulated within the β–CD cavity, the free cationic peptide initially interacts with the negatively charged bacterial membrane. Meanwhile, the hydrophilic region points toward the inner side of the cell membrane, creating membrane perforations. This mechanism targeting the bacterial cell membrane is less prone to inducing drug resistance, making it an ideal candidate for combating pathogens, either independently or in combination with antibiotics.

As expected, the biocompatibility of LA–modified peptides has been greatly improved by β–CD inclusion. Engineered AMPs typically have a hydrophobic region, which can bind to lipids and induce hemolysis. On the other hand, although a benzylamine modification at the C–terminus can significantly enhance antibacterial performance, its biocompatibility is relatively poor. The hemolysis rate of LRP is approximately twice that of LR at higher concentrations. However, after inclusion with β–CD, the hemolysis rates of LR and LRP decrease to half or even lower at the same concentration. The cytotoxicity of β–CD/LR is reduced at higher concentrations, while that of β–CD/LRP is significantly reduced at various concentrations. The β–CD inclusion provides protection for the hydrophobic region without affecting the antibacterial performance. Animal experiment results further confirmed the antibacterial effects of the inclusion complexes in vivo. In the infected mice, no major inflammatory response occurred when treated with the inclusion complexes. The safety profile is significantly important for their further application in biomedical areas.

## 4. Materials and Methods

### 4.1. Materials

The test strains *Escherichia coli* ATCC 25922 (*E. coli*), *Escherichia coli* DC2 (*E. coli* DC2), *Staphylococcus aureus* ATCC 29213 (*S. aureus*), *Candida albicans* ATCC 10231(*C. albicans*), and *methicillin–resistant Staphylococcus aureus* ATCC 43300 (*MRSA*) were obtained from Shanghai Bioresource Collection Center, SHBCC (Shanghai, China). Luria–Bertani (LB) broth, trypticase soy broth (TSB), Sabouraud dextrose broth, and Mueller–Hinton broth (MHB) were purchased from Hopebiol (Qingdao, China). Alpha–lipoic acid (LA), β–cyclodextrin (β–CD), phenethylamine (PEA), methanol, glacial ether, 3,3–dipropylthiodicarbocyanine iodide (diSC_3_5), N–phenyl–1–naphthylamine (NPN), crystal violet, dimethyl sulfoxide (DMSO), methyl–thiazolyl diphenyl–tetrazolium bromide (MTT), liquid paraffin, hematoxylin–eosin dye solution, neutral gum and other chemicals were purchased from Aladdin Co. (Shanghai, China). Mouse fibroblast cells (L929 cells) and newborn calf serum were obtained from Key–GEN BioTECH (Nanjing, China). BALB/C mice were purchased from the Qinglongshan Animal Breeding Farm (Nanjing, China).

### 4.2. Peptide Synthesis and Purification

The antimicrobial peptide (LR) consisting of arginine (R), tryptophan (W), and a–arginine (R) and modified with alpha lipoic acid (LA) was synthesized using an Fmoc solid–phase synthesis method. The 2–chlo–rotrityl resin was subjected to a peptide synthesizer and extended using a carbodiimide–based activation following an initial Fmoc–deprotection. LA was treated as the fourth amino acid. After being cleaved from the resin, LR was modified by PEA (LRP) on the C–terminus via DIC/HOBt coupling. The final products were purified using reverse–phase high–performance liquid chromatography (RP–HPLC). All lipopeptides were analyzed and confirmed via HPLC and electrospray ionization mass spectrometry (ESI–MS) (see Figure A1, Figure A2, Figure A3 and Figure A4 in Appendix A for details). The molecular ion peaks of LR and LRP were 704.3/808.4, which were consistent with the theoretical molecular weight, and the purities were 90.4%/90.7%.

### 4.3. Preparation and Characterization of β–CD/LR and β–CD/LRP Inclusion Complexes

Firstly, β–CD and LR were prepared as different mass ratios; β–CD and LRP were also prepared as different ratios. These mixtures were sonicated for 3 h using an ultrasonic cleaner (PS10–250A) to demonstrate the occurrence of encapsulation (shown in Figure 9) using Fourier infrared spectroscopy (Nicolet 5700) and using a UV–visible spectrophotometer (U–3900) to determine the best solution ratios. The β–CD/LRP was dried in a freeze–dryer overnight and then observed via transmission electron microscopy (TEM, JEOLJEM 2100). Each sample was repeated at least three times, and all experiments were performed at room temperature.

### 4.4. Minimum Inhibitory Concentration (MIC) and Minimum Bactericidal Concentration (MBC) Measurements

The inhibitory and bactericidal activities of β–CD/LR and β–CD/LRP were tested by measuring the minimum inhibitory concentration (MIC) and the minimum bactericidal concentration (MBC) [28], using LR, LRP, and the commercial daptomycin as controls. Aqueous stock solutions of β–CD/LR, β–CD/LRP, LR, and LRP were diluted stepwise from 512 to 1 μg/mL with MHB. Each dilution was transferred to a microtiter plate, followed by the addition of the bacterial suspension (bacterial concentration of 1 × 10^5^ CFU/mL). MHB without the encapsulated peptide antibacterial formulation was used as a control, and the MICs were quantified by measuring the optical density at 540 nm. The MBCs were measured by aspirating the solution from each clear well (≥MIC) onto an agar plate. The MBCs were determined to be the lowest concentration, around 99% bactericidal. All experiments were performed in triplicate.

### 4.5. Stability 

To study the stability of LR, LRP, β–CD/LR, and β–CD/LRP against *E. coli* under different pH (pH 6.8, pH 7.4, and pH 8.0), salt (100 mM NaCl and 1 mM CaCl_2_), serum (10% and 20%), and time (30 days later) conditions, the MICs were determined under different conditions. Each experiment was repeated three times at room temperature.

### 4.6. Planktonic Killing Kinetics

LR, β–CD/LR, LRP, and β–CD/LRP were each added to 2 mL of the *E. coli* suspension at a final concentration of 2MIC. The mixture was then shaken at 200 rpm at 37 °C. Time–points selected to assess planktonic kill kinetics were 5 min, 30 min, 1 h, 2 h, and 4 h. At each predetermined time point, 100 μL of the *E. coli* suspension was spread onto the surface of agar plates. After incubation overnight, the viable cell colonies were counted and expressed as the mean colony units per milliliter (CFU/mL). The same procedure was followed for the other bacteria, and each experiment was repeated three times.

### 4.7. Anti–Biofilm Assay

Initially, 1 mL of *E. coli* was introduced into each well of a 48–well plate, followed by the addition of 1 mL of sterile PBS to the surrounding wells. The plate was incubated at 37 °C for 5 h to facilitate the formation of a bacterial biofilm. Subsequently, 1 mL of LR, β–CD/LR, LRP, and β–CD/LRP (at 4MIC) was added to the respective wells. One well containing 1 mL of medium served as the growth control group, while another containing PBS was designated as the blank control group. Additionally, a control group using daptomycin was included.

Then, the medium was aspirated and discarded and rinsed with PBS, after which 1 mL of methanol was added to each well, and the methanol was fixed for 15 min and then discarded and air–dried. Next, 1 mL of 0.1% crystalline violet was added to each well. Each well was stained with 1 mL of 0.1% crystalline violet solution and rinsed with PBS to remove the floating color. Finally, 1 mL of 33% acetic acid solution was added, and the absorbance was measured at 590 nm using a microplate reader, with higher absorbance values indicating a higher bacterial biofilm content [29]. The formula was as follows: film killing rate (%) = [(absorbance value of experimental group − absorbance value of blank group)/(absorbance value of growth control group − absorbance value of blank group)] × 100%. 

### 4.8. Visualization of Cell Damage

The *E. coli* suspension was added to the confocal Petri dish and incubated statically at 37 °C for 48 h. The supernatant medium was removed and aspirated and rinsed with PBS. Then, 1 mL of LR, β–CD/LR, LRP, and β–CD/LRP (2MIC) and 1 mL of LB medium as a blank control were added, respectively, and incubation continued for 5 h at 37 °C. Next, 500 μL of mixed fluorescent dye was added, incubated for 30 min, and rinsed with PBS, and then, it was incubated with a laser confocal microscope (CLSM, 4200A–scs) with excitation. *C. albicans* bacteria were characterized at a wavelength of 488 nm. The experiments were repeated three times.

### 4.9. Study of Antibacterial Mechanism

#### 4.9.1. Depolarization

The depolarizing effect of β–CD/LR and β–CD/LRP on bacterial cell membranes was detected using diSC_3_5, a fluorescent probe sensitive to the membrane potential. Overnight cultures of *E. coli* were collected, washed twice with HEPES, diluted to 5 × 10^7^ CFU/mL, mixed with a 1.83 Mm diSC_3_5 solution, and incubated for 30 min. After that, KCl solution was added to the bacterial suspension and incubated for 10 min. Then, β–CD/LR and β–CD/LRP (2MIC) were added to the mixture, and the fluorescence emission spectra of the solution (excitation wavelength of 622 nm, emission wavelength of 670 nm) were measured immediately after mixing, using a fluorescence spectrophotometer (F–7000). The experiments were all repeated three times at room temperature.

#### 4.9.2. Extracellular Membrane Permeability

The extracellular membrane permeation of β–CD/LR and β–CD/LRP was detected using the fluorescent probe NPN. *E. coli* was washed with PBS and diluted to 5 × 10^7^ CFU/mL. Then, 10 mL of the diluted bacterial solution was taken, NPN solution was added to a final concentration of 10 μM, β–CD/LR and β–CD/LRP (2MIC) were added to the mixture, and the fluorescence emission spectra of the solutions (excitation wavelength of 350 nm, emission wavelength of 429 nm) were measured immediately after mixing on a fluorescence spectrophotometer. The experiments were all repeated three times at room temperature.

### 4.10. Study of Biocompatibility

#### 4.10.1. Hemolysis of HRBCs 

The erythrocytes was prepared into a 2% (*v*/*v*) suspension with saline. Three negative controls (with saline) and three positive controls (with distilled water) were set up; 2.5 mL of 2% erythrocyte suspension and 100 μL of LR, β–CD/LR, LRP, and β–CD/LRP were added to each centrifuge tube, followed by 2.4 mL of saline, and then gently mixed and incubated in an incubator at 37 °C for 3–4 h. The tubes were removed and centrifuged for 5 min. The absorbance of each well was measured at 545 nm using a microplate reader (Model 680, Bio–Rad, Hercules, CA, USA), with saline as a control [30]. The hemolysis rate was calculated by taking the average value (hemolysis was considered as hemolysis when the hemolysis rate was >5%), and the formula was as follows: hemolysis rate (%) = [(absorbance value of the experimental group—absorbance value of the negative control group)/(absorbance value of the positive control group—absorbance value of the negative control group)] × 100%. The experiments were repeated three times at room temperature.

#### 4.10.2. In Vitro Cytotoxicity

L929 cells at a logarithmic growth stage were spread in 96–well plates at a density of 8 × 10^3^ cells/well and incubated in an incubator (5% CO_2_, 37 °C) for 24 h. After that, LR, β–CD/LR, LRP, and β–CD/LRP were added; the one without the antimicrobial peptide was used as the growth control group, and the one with PBS was used as the blank control group. MTT solution was added and incubated for 4 h, and the supernatant was aspirated and discarded. Then, 100 μL DMSO was added to each well, and the absorbance value was detected at 490 nm using a microplate reader. The formula was as follows: cell survival rate (%) = [(absorbance value of experimental group − absorbance value of blank group)/(absorbance value of growth control group − absorbance value of blank group)] × 100%. The experiments were repeated three times.

### 4.11. In Vivo Experiment

Three–to–five–week–old female *BALB/C* mice were purchased from the Qinglongshan Animal Breeding Farm (Nanjing, China), weighing 20–25 g. The animals were kept under standard conditions with free access to food and water. All studies were performed in accordance with the Guide for Care and Use of Laboratory Animals, as proposed by the Committee of Southeast University, China.

#### 4.11.1. In Vivo Antibacterial Kinetics

*BALB/C* mice were intravenously injected with 300 μL of *MRSA* bacteria at a concentration of 1 × 10^8^ CFU/mL. Injections of 80 μL of LR (or β–CD/LR) at a concentration of 6.67 mg/mL or 100 μL of LRP (or β–CD/LRP) at 1.67 mg/mL were administered after 24 h. At 27 h, 30 h, 36 h, and 48 h, mice were subjected to orbital blood sampling, respectively. A 100 μL blood sample was spread on the plates and subsequently incubated at 37 °C overnight for bacterial counting. Three mice were set up as the control group, with twelve mice as the experimental group.

#### 4.11.2. Tissue Sections

After 48 h, the livers and spleens of the mice were removed and embedded with liquid paraffin, and the tissues were then sectioned with an ultrathin sectioning machine (UC7), with the one without bacteria as the negative control group and the one without the inclusion complexes as the positive control group. The sections were placed in a dyeing vat with hematoxylin for 3 min, rinsed well with tap water, and observed under a microscope to show dark purple nuclei and a light blue cytoplasm. The sections were then placed in a 1% hydrochloric acid alcohol solution for 3 s and taken out quickly, and the nuclei were observed to be purple–blue, and the cell pulp was colorless under the microscope; it was then washed with tap water, and then, counter–blue with 1% ammonia was added until the nuclei were blue, and it was washed with tap water. The sections were placed in eosin dye solution for 1 min, rinsed in tap water, and sealed with neutral gum, and the liver and spleen tissues were viewed and recorded under a microscope.

## 5. Conclusions

Based on the host–guest interaction between β–CD and LA, engineered tripeptides were successfully encapsulated to form nano–inclusion complexes. Both β–CD/LR and β–CD/LRP exhibited a significant improvement in antibacterial performance against *E. coli*, *S. aureus*, *C. albicans*, and *MRSA*. β–CD/LR inclusion complexes indicated an approximately 32–fold decrease in MICs/MBCs against *E. coli* and *S. aureus*. β–CD/LRP inclusion complexes achieved a biofilm removal rate of 87.7%, comparable to that of commercially available daptomycin. Depolarization and outer membrane permeability results demonstrated the swift disruption of the bacterial membrane with both inclusion complexes. Moreover, the hemolysis rate and cytotoxicity of the inclusion complexes were significantly reduced, with a more notable reduction at high concentrations. In addition to its biosafety profile, the in vivo bactericidal activity of β–CD/LRP was superior, demonstrating a bactericidal efficiency exceeding 99.99% within 6 h. β–CD encapsulation was proven to be an effective strategy for optimizing engineered peptides with excellent antibacterial performance and favorable biocompatibility. Constructing nano–inclusion complexes would further extend the applications of engineered AMPs in biomedical areas.

## Figures and Tables

**Figure 1 ijms-24-14801-f001:**
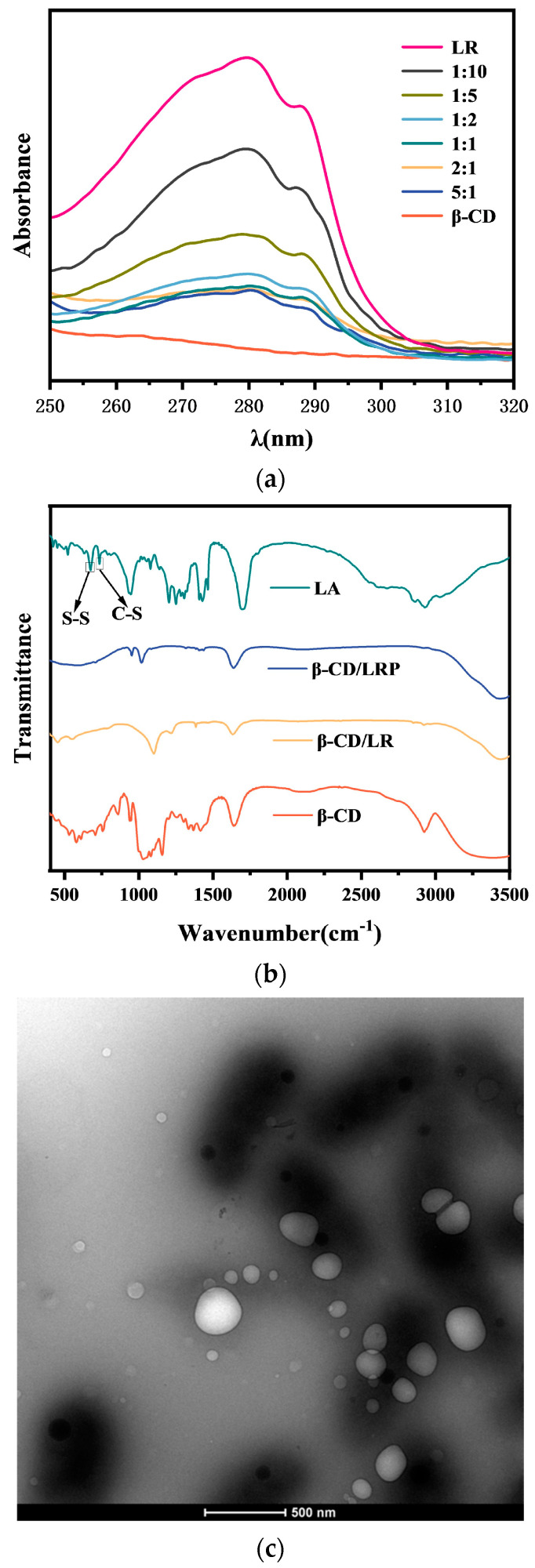
Characterization plots of inclusion complexes: (**a**) UV spectrogram of β–CD/LR; (**b**) IR spectra of β–CD/LR at 1:5 and β–CD/LRP at 1:3; (**c**) TEM plots of β–CD/LRP at 1:3.

**Figure 2 ijms-24-14801-f002:**
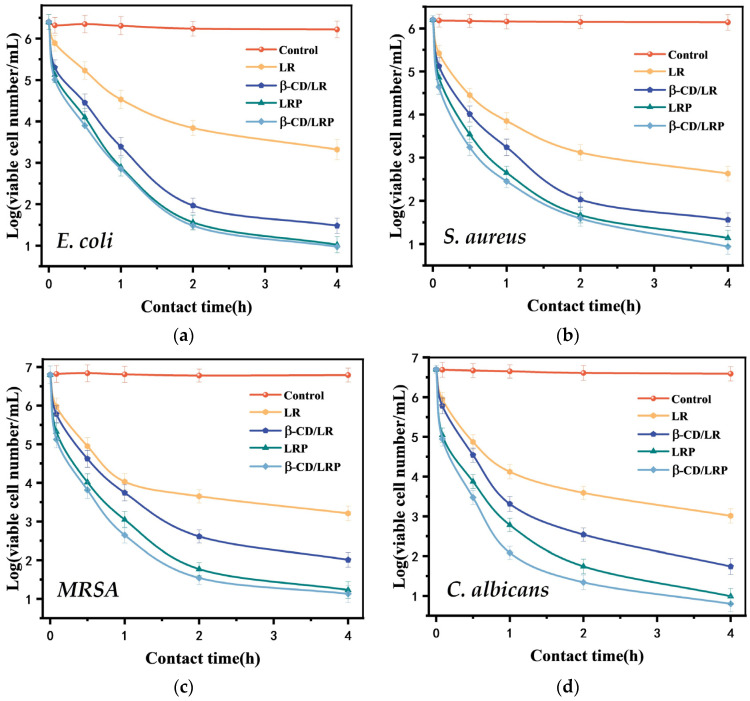
Time–kill curves of LR, β–CD/LR, LRP, and β–CD/LRP (2MIC) against (**a**) *E. coli*, (**b**) *S. aureus*, (**c**) *MRSA*, and (**d**) *C. albicans*.

**Figure 3 ijms-24-14801-f003:**
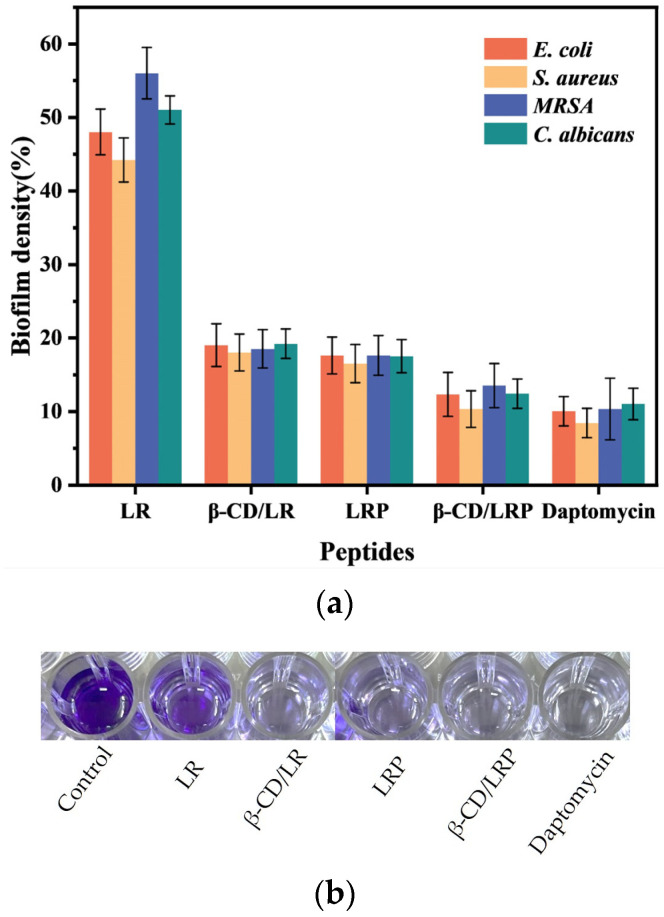
(**a**) Effects of LR, β–CD/LR, LRP, and β–CD/LRP (4MIC) on the biofilm activities of *E. coli*, *S. aureus*, *MRSA*, and *C. albicans*. (**b**) The biofilm content of *E. coli* determined based on the crystalline violet method. Bacteria were grown in medium to a concentration of 1 × 10^7^ CFU/mL, placed in 24–well plates, and incubated at 37 °C for 48 h to form biofilms.

**Figure 4 ijms-24-14801-f004:**
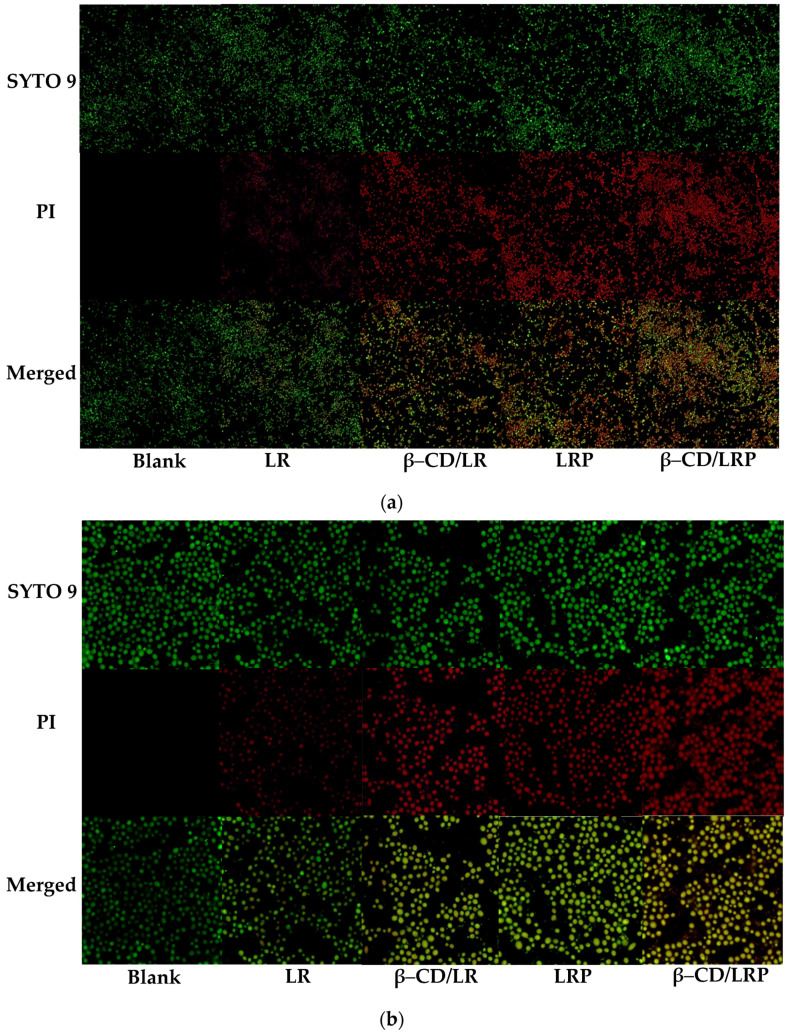
CLSM images of (**a**) *E. coli* and (**b**) *C. albicans* biofilms treated with LR, β–CD/LR, LRP, and β–CD/LRP at a 2 MIC concentration for 5 h, respectively.

**Figure 5 ijms-24-14801-f005:**
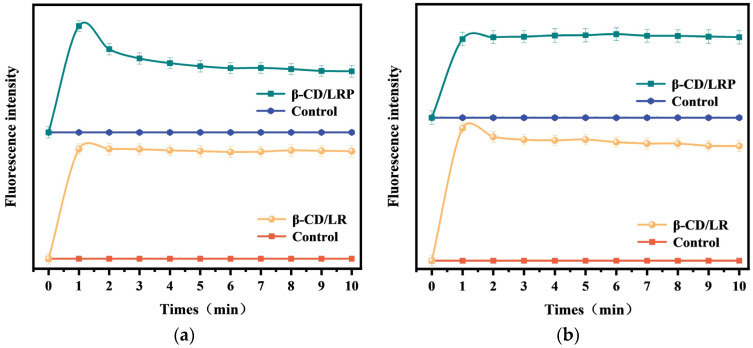
(**a**) Cytoplasmic membrane depolarization and (**b**) outer membrane permeability of *E. coli* treated with β–CD/LR and β–CD/LRP at the concentration of 2MIC for 10 min, respectively.

**Figure 6 ijms-24-14801-f006:**
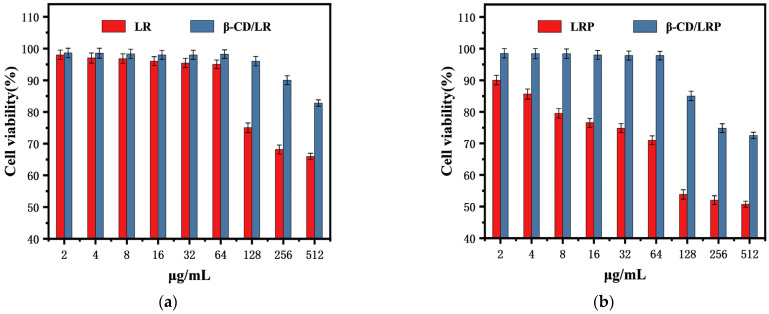
The toxic effects of (**a**) LR and β–CD/LR and (**b**) LRP and β–CD/LRP on L929 cells at different concentrations, with the cell density at 1 × 10^6^ CFU/mL.

**Figure 7 ijms-24-14801-f007:**
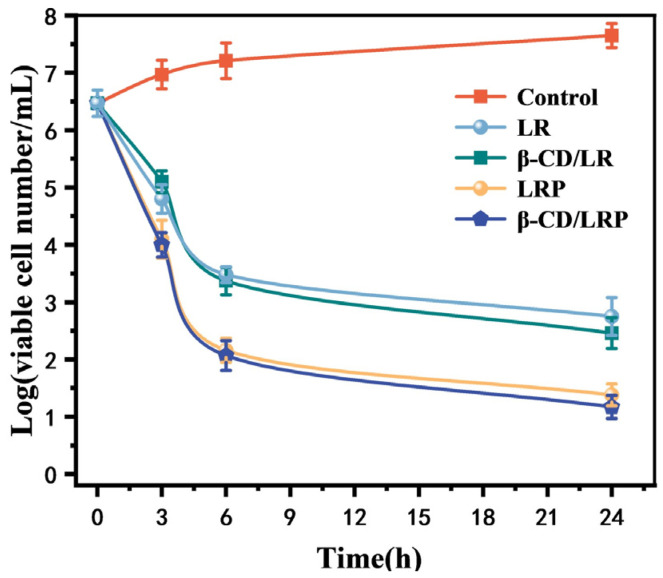
In vivo antibacterial kinetics in mice.

**Figure 8 ijms-24-14801-f008:**
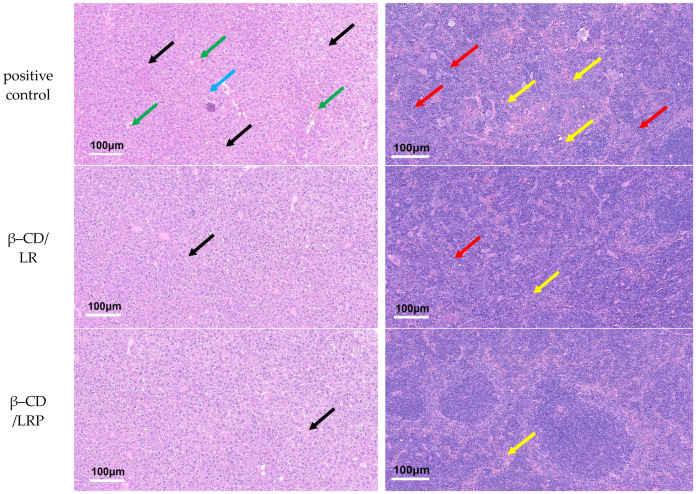
Tissue sections of mice (HE staining): (**a**) liver: cytoplasmic laxity and light staining (black arrow), lymphocytic punctate infiltration (blue arrow), central venous stasis (green arrow); (**b**) spleen: irregular shape of the white pith (red arrow), neutrophil infiltration (yellow arrow).

**Figure 9 ijms-24-14801-f009:**
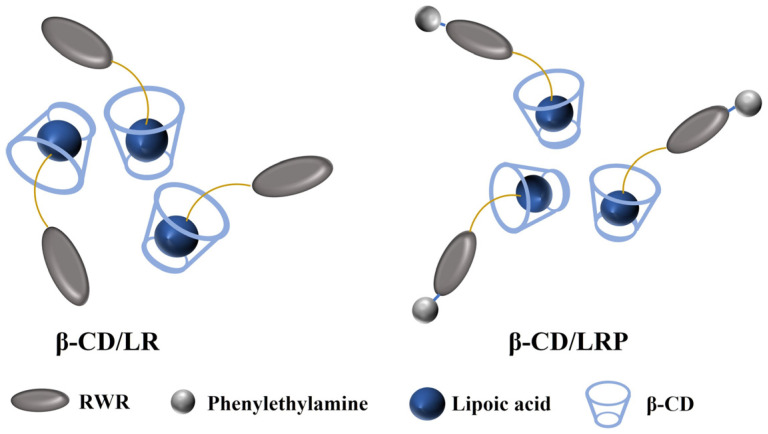
Synthesis route of inclusion complexes.

**Table 1 ijms-24-14801-t001:** The MICs and MBCs of LR, β–CD/LR, LRP, β–CD/LRP, and daptomycin as a control.

Peptides (μg/mL)	*E. coli*		*S. aureus*		*MRSA*		*Candida albicans*	
	MIC	MBC	MIC	MBC	MIC	MBC	MIC	MBC
LA–RWR(LR)	128	256	256	512	256	256	128	256
β–CD/LR	4	8	4	4	8	16	8	8
LA–RWR–PEA(LRP)	4	8	4	8	8	16	8	16
β–CD/LRP	2	4	4	8	4	8	4	8
Daptomycin	1	2	1	2	2	4	2	2

**Table 2 ijms-24-14801-t002:** The MICs of LR, LRP, β–CD/LR, and β–CD/LRP against *E. coli* in different environments.

Peptides	pH 6.8	pH 7.4	pH 8	100 mM NaCl	1 mM CaCl_2_	10% Serum	20% Serum	30 Days
β–CD/LR	4	4	4	8	8	4	8	4
β–CD/LRP	2	2	2	2	4	2	4	2

**Table 3 ijms-24-14801-t003:** Hemolysis rates of LR, β–CD/LR, LRP, β–CD/LRP, and daptomycin as a control.

Peptides (μg/mL)	Hemolysis Rate (%)									
	1	2	4	8	16	32	64	128	256	512
LR	2.02	2.09	2.14	2.22	2.33	2.47	2.66	2.95	3.26	5.37
β–CD/LR	0.86	1.09	1.11	1.13	1.16	1.19	1.27	1.39	1.78	2.14
LRP	0.55	0.57	1.25	2.27	2.84	3.17	3.98	4.41	4.85	5.77
β–CD/LRP	0.31	0.42	1.14	1.43	1.76	2.06	2.07	2.18	2.35	2.45
Daptomycin	1.04	1.08	1.15	1.72	1.98	2.36	3.04	3.32	3.97	4.87

## Data Availability

Not applicable.

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
