# Peer review of "Optimization of Antibacterial Activity and Biosafety through Ultrashort Peptide/Cyclodextrin Inclusion Complexes"

_ijms, 2023, doi:10.3390/ijms241914801_

Round 1
Reviewer 1 Report
This paper describes the use of beta-cyclodextrin to improve antimicrobial activity of a short, cationic peptide while increasing biocompatibility with eukaryotic cells. Overall, the work is interesting and should be published. However, there are many issues to be addressed and questions to be answered before publication should be considered. These are detailed below.
1. There are too many grammatical and other usage mistakes to list. Examples: non-matched noun/verb plurals, mis-used articles, sentences started with numbers. Careful editing must be done before publication.
2. The authors state that specific cyclodextrin-peptide ratios are “best.” No indication is given for the criteria used for this judgement. One UV titration is provided, which presumably involves a change in tryptophan absorbance upon association with the cyclodextrin. From these data, an association constant can be (should be) measured and reported. And the rationale for use of specific ratios of peptide and cyclodextrin must be provided. These things should be done with both peptides.
3. The TEM images (not “plots”) shown in Figure 2 are meaningless without some additional explanation. The structures observed have sizes above 100 nm and therefore must be aggregates of some form or another. No explanation is given regarding these structures.
4. The authors perform membrane depolarization experiments with E. coli but their descriptions are not accurate. The authors claim that “depolarization can be visualized by the release of the fluorescent probe…” The increase in fluorescence is due to depolarization of the inner membrane. They observe an initial drop in fluorescence, which they explain as “structural collapse…” It is not clear what is meant by this term. These depolarization experiments require that the probe (diSC35) gain access to the cytoplasmic membrane. The outer membrane of Gram-negative bacteria inhibits this access. Over time, some of the probe will diffuse through the outer membrane. Addition of their antimicrobial will necessarily permeabilize the outer membrane, allowing the probe to insert into the polarized cytoplasmic membrane, which will decrease its fluorescence.
5. The authors claim to administer nearly 10^8 CFU of MRSA IV into mice. This inoculum is 10 to 100-fold higher than is typically used for bacteremia models in mice (for example see: J Immunol Methods. 2014 Aug; 410: 88–99.). The inoculum described should be lethal within the 24 h incubation time.
There are too many grammatical and other usage mistakes to list. Examples: non-matched noun/verb plurals, mis-used articles, sentences started with numbers. Careful editing must be done before publication.
Reviewer 2 Report
In this manuscript, Hang Liu, Lin Wang and Chen Yao presented very interesting and important studies, in which they showed the possibility of using cationic peptides as an alternative to natural antimicrobial peptides. To enhance the antibacterial activity, the authors created complexes of the mentioned peptides with beta cyclodextrins. This work is a novelty despite many previously published works describing the use of cyclodextrins to improve the effectiveness of some drugs However, the great advantage of this work is the possibility of a significant reduction of the minimal inhibitory concentration of the peptides used to combat bacteria. Thanks to this, it is possible to reduce cytotoxicity and improve the effectiveness of treatment. In general, the work is well written and understandable and arouses the interest of the reader. However, this work requires minor corrections and clarifications:
1. Section 2.3. Preparation and Characterization of β-CD/LR and β-CD/LRP Inclusion Complexes. The authors presented one specific time (3 hours) of using ultrasound to form complexes with cyclodextrins. Was only one sonication time carried out? How do you know that this particular time was sufficient? What was the percentage of peptide-CD complexes before and after three hours?
2. Section 3. Results (line 251-253) and figure 2a. Why, looking at the changing spectra, was it found that the best complexation occurs in the proportions 1:5? A very short explanation would be useful here.
3. Figure 1 shows that the anchoring of the RWR molecules (antimicrobial peptide (LR) consisting of arginine (R), tryptophan (W) and α-arginine 10 (R) and modified with alpha lipoic acid (LA)) occurs as a result of interaction LA with CD structure. The figure shows that they are combined in a 1:1 ratio. The results described in Section 3 (Results) show that they combine in a ratio of 1:5. Does each B-CD molecule bound to as many as 5 LR molecules? A short explanation would be useful.
4. How was the stability of β-CD/LR and β-CD/LRP complexes tested? Only by comparing its MICs under different conditions?
I have no particular reservations about the quality of the language.
